# LANGUAGE MODEL PRE-TRAINING IMPROVES GENERALIZATION IN POLICY LEARNING

## ABSTRACT

Language model (LM) pre-training has proven useful for a wide variety of language processing tasks, including tasks that require nontrivial planning and reasoning capabilities. Can these capabilities be leveraged for more general machine learning problems? We investigate the effectiveness of LM pretraining to scaffold learning and generalization in autonomous decision-making. We use a pre-trained GPT-2 LM to initialize an interactive policy, which we fine-tune via imitation learning to perform interactive tasks in a simulated household environment featuring partial observability, large action spaces, and long time horizons. To leverage pre-training, we first encode observations, goals, and history information as templated English strings, and train the policy to predict the next action. We find that this form of pre-training enables generalization in policy learning: for test tasks involving novel goals or environment states, initializing policies with language models improves task completion rates by nearly 20%. Additional experiments explore the role of language-based encodings in these results; we find that it is possible to train a simple adapter layer that maps from observations and action histories to LM embeddings, and thus that language modeling provides an effective initializer even for tasks with no language as input or output. Together, these results suggest that language modeling induces representations that are useful for modeling not just language, but natural goals and plans; these representations can aid learning and generalization even outside of language processing.

## 1 INTRODUCTION

In recent years, **language models** (LMs) trained on open-domain text corpora have come to play a central role in machine learning approaches to natural language processing tasks (Devlin et al., 2018). This includes tasks that are not purely linguistic, and additionally require nontrivial planning and reasoning capabilities: examples include as vision-language navigation (Majumdar et al., 2020; Fried et al., 2018; Suglia et al., 2021), instruction following (Zhang & Chai, 2021; Hill et al., 2020), and visual question answering (Tsimpoukelli et al., 2021). Indeed, some of these tasks are so remotely connected to language modeling that it is natural to ask whether the capabilities that result from LM pre-training might extend to tasks that involve no language at all—and if so, how these capabilities might be accessed in a model trained only to process and generate natural language strings. In this paper, we study these questions through the lens of **embodied decision-making**, investigating the effectiveness of LM pretraining as a scaffold for learning control policies for interactive tasks featuring partial observability, large action spaces, complex states, and complex dynamics. We describe a series of experiments in the VirtualHome environment (Puig et al., 2018; 2020) in which LMs are used to initialize policies, and show that LM pre-training substantially improves generalization across common-place tasks in household environments.

In **Experiment 1** (Section 6), we encode the inputs to a policy—including observations, goals, and action histories—as templated English phrases (e.g. representing the goal on(fork, table) as *There is a fork on the table.*) as shown in Figure 1. A pretrained LM is then fined-tuned to produce representations of these phrases that can be used to predict subsequent actions. For i.i.d. training and evaluation tasks, we find that this approach completes tasks at a rate comparable to the same transformer-based policy trained from scratch. For generalization to out-of-distribution tasks, however, LM pretraining confers substantial benefits: it improves task completion rates by

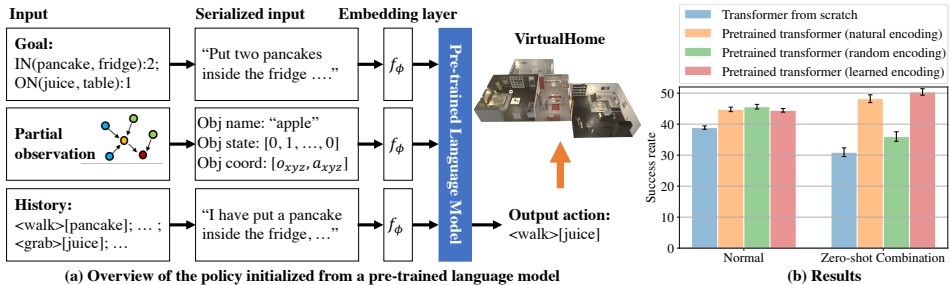

(a) Overview of the policy initialized from a pre-trained language model

(b) Results

Figure 1: **(a) Overview of the proposed method.** We encode the inputs to the policy — including observations, goals, and action histories — as templated English phrases. The phrases are sent to an embedding layer and a pre-trained language model to predict the subsequent action. The embedding layer $f_\phi$ can either be a embedding layer of a pre-trained language model (Experiment 1) or be a learned embedding layer (Experiment 2). **(b) Results.** We summarize the main results of Experiment 1 ("Transformer from scratch" and "Pretrained transformer (natural encoding)"), Experiment 2A ("Pretrained transformer (random encoding)") and Experiment 2B ("Pretrained transformer (learned encoding)").

nearly 20% for tasks involving novel initial environment states and goals (Figure 1 "Transformer from scratch" and "Pretrained transformer (natural encoding)" ).

Next, we conduct two experiments aimed at clarifying the role of this string-based encoding. We design Experiment 2A that uses random strings instead of natural language inputs and Experiment 2B that uses non string-based encodings to study different ways of building interfaces between input encodings and LMs. In **Experiment 2A** (Section 7), we replace the "natural" string encodings of Experiment 1 with an arbitrary mapping between logical goals and tokens (e.g. serializing on(fork, table) as *brought wise character trees fine order yet*). This random encoding substantially (by roughly 12%) degrades performance on out-of-distribution tasks, indicating that LM encoders are sensitive to the form of string encodings even when fine-tuned (Figure 1 "Pretrained transformer (natural encoding)" and "Pretrained transformer (random encoding)"). In **Experiment 2B**, we investigate whether string-based encodings are necessary at all. We replace the model's word embedding layer with a randomly initialized embedding layer that maps from discretized environment observations, goal, and history actions to a sequence of LM input vectors, and fine-tune this embedding layer jointly with the LM itself. This learned encoder performs almost the same as the encoding of Experiment 1, indicating that effective encodings for non-linguistic tasks can be learned from scratch (Figure 1 "Pretrained transformer (natural encoding)" and "Pretrained transformer (learned encoding)").

These experiments offer two main conclusions. First, they show that **language modeling improves generalization in policy learning**: initializing a policy with a neural LM (pre-trained on a next-word prediction task with a large text corpus) substantially improves out-of-distribution performance on (non-linguistic) tasks in an interactive environment. Second, they show that **language-based environment encodings are not needed to benefit from LM pretraining**: it is instead possible to learn an interface between observations, actions, and model-internal representations derived from text corpora. These results point the possible effectiveness of **language modeling as a general-purpose pre-training scheme** to promote structured generalization in broader machine learning applications.

## 2 RELATED WORK

In recent years, word and sentence representations from pre-trained LMs (Peters et al., 2018; Devlin et al., 2018; Radford et al., 2018) have become ubiquitious in natural language processing, playing a key role in state-of-the-art models for tasks ranging from machine translation (Zhu et al., 2020) to task-oriented dialog (Platanios et al., 2021). Some of the most successful applications of pre-training lie at the boundary of natural language processing and other domains, as in instruction following (Hill et al., 2020) and language-guided image retrieval (Lu et al., 2019). Building on this past work, our experiments in this paper aim to explain whether these successes result entirely from improved processing of text, or instead from domain-general representational abilities. Below, we briefly survey existing applications of pretraining that motivate the current study.

**Learning representations of language** From nearly the earliest days of the field, natural language processing researchers have observed that representations of words derived from distributional statis-

tics in large text corpora serve as useful features for downstream tasks (Deerwester et al., 1990; Dumais, 2004). The earliest versions of these representation learning schemes focused on isolated word forms (Mikolov et al., 2013; Pennington et al., 2014). However, recent years have seen a number of techniques for training (masked or autoregressive) language models to produce contextualized word representations (which incorporate information neighboring words in sentences and paragraphs) via a variety of masked-word prediction objectives (Devlin et al., 2018; Yang et al., 2019).

**Applications** In addition to producing useful representations, these language models can be fine-tuned to perform language processing tasks other than language modeling by casting those tasks as word-prediction problems. Successful uses of representations from pretrained models include syntactic parsing (Kitaev et al., 2018) and language-to-code translation (Wang et al., 2019); successful adaptations of LM prediction heads include machine translation (Zhu et al., 2020), sentiment classification (Brown et al., 2020) and style transfer (Keskar et al., 2019). Text-based games (Yao et al., 2021; Yuan et al., 2018; Ammanabrolu & Riedl, 2018; Côté et al., 2018) inherently involve text as both the input and the output. Recent works (Yao et al., 2020) in text-based games use GPT-2 to solve the text-based games and get significant performance improvements. However, it is hard to describe 3D information using text and most of their experiments are in 2D environments. Included in these successes are a number of tasks that integrate language and other modalities, including visual question answering and image captioning (Yang et al., 2020). In models that condition on both text and image data, several previous approaches have found that image representations can be injected directly into language models' embedding layers (Tsimpoukelli et al., 2021) using a similar mechanism to the one we describe in Experiment 2B. One of our main contributions is to show that approach works even for tasks in which *only* non-linguistic information is relevant to model predictions.

**What do LMs encode?** The possibility that LMs might encode non-linguistic information useful for other downstream tasks is suggested by a number of recent "probing" studies aimed at understanding their predictions and the structure of their internal representations. Pre-trained LMs can answer a non-trivial fraction of queries about both factual and common-sense knowledge (Roberts et al., 2020). Their representations encode information about perceptual relations among concepts, including visual similarity among object classes (Ilharco et al., 2020) and the structure of color spaces (Abdou et al., 2021). Finally, they appear to be capable of basic simulation, modeling changes in entity states and relations described by text (Li et al., 2021).

**LM pretraining beyond language** Two recent papers consider questions closely related to the ones investigated here: (Brown et al., 2020) show that the GPT-3 model is capable of performing a limited set of arithmetic and string manipulation tasks; (Lu et al., 2021) show that pretrained LMs require very little fine-tuning to *match* the performance of task-specific models on several image classification and numerical sequence processing tasks. In this paper, we focus on non-linguistic tasks where the inputs and outputs do not involve language. To the best of our knowledge, the current study is the first to demonstrate improved generalization in a non-linguistic problem over a standard neural-network baseline using a pre-trained language model.

## 3 LANGUAGE MODELING AND POLICY LEARNING

We begin with a brief review of the ingredients of language modeling and policy learning tasks used in our experiments.

### 3.1 LANGUAGE MODELING

Our experiments in this paper focus on **autoregressive**, **transformer-based** language models (Vaswani et al., 2017). These models are trained to fit a distribution $p_\theta(\boldsymbol{y})$ over a text sequence $\boldsymbol{y}$ by decomposing it into a sequence of tokens $\boldsymbol{y} = \{y_1, y_2, \ldots, y_n\}$ via the chain rule:

$$\log p_\theta(\boldsymbol{y}) = \sum_{i=1}^{n} \log p_\theta(y_i \mid y_1, y_2, \ldots, y_{i-1}). \tag{1}$$

Each conditional distribution $p_\theta(y_i|y_1, y_2, \ldots, y_{i-1})$ is parameterized by a transformer neural network $f_\theta(y_1, y_2, \ldots, y_{i-1})$. This network encodes each conditioned token $y_i$ into a continuous embedding $e_i = g(y_i)$ which is then fed into the transformer architecture and encoded into a categorical distribution over token values of $y_i$. Our experiments utilize a standard language model, GPT-2, that is trained on Webtext dataset (Radford et al., 2018) using Huggingface library (Wolf et al., 2019).

## 3.2 POMDPs and Policy Learning

Our experiments explore the application of LMs to general sequential decision-making tasks in partially observed environments. These tasks may be formalized as partially observable Markov decision processes (POMDPs). A POMDP is defined by a set of states $\mathcal{S}$, a set of observations $\mathcal{O}$, a set of actions $\mathcal{A}$, and a transition model $\mathcal{T}(s_{t+1}|s_t, a_t)$ that predicts the next state $s_{t+1}$ based on the current state $s_t$ and an action $a_t$. Importantly, in a POMDP setting, the observation $o_t$ only captures a portion of the underlying state $s_t$, and an optimal decision-making strategy (a **policy**) must incorporate both the current observation and the previous history of observations and actions. For experiments in this paper, policies are parametric models $\pi_\psi(a_t|g, h, o_t)$ that select actions given the goals $g$, history information $h$, and partial observations $o_t$ of the current state $s_t$.

All our experiments use imitation learning (Santara et al., 2017; Ng et al., 2000; Peng et al., 2018), specifically behavior cloning (Pomerleau, 1991; 1989; Torabi et al., 2018), to train $\pi_\psi$. We collect a dataset of $\hat{N}$ expert training trajectories $\mathcal{D} = \{d_1, \cdots, d_{\hat{N}}\}$, where each individual trajectory consists of a set of goal, observations, and actions, *i.e.* $d_i = \{o_1, a_1, \cdots, a_T, g\}$, where $T$ is the length of an expert trajectory. We then train a policy $\pi_\psi(a_t|g, h_t, o_t)$ which maximizes the likelihood $p_\psi(\boldsymbol{a})$ of the expert actions $\boldsymbol{a} = \{a_1, \cdots, a_T\}$ taken in a trajectory using supervised learning,

$$\log p_\psi(\boldsymbol{a}) = \sum_{t=1}^{T} \log p_\psi(a_t \mid g, h_t, o_t), \tag{2}$$

where $h_t$ consists of all history in the environment up to timestep $t$.

## 3.3 Language models as policy initializers

Ultimately, both language modeling and POMDP decision-making involve a sequence predictions (over words or actions) given a sequence of previous observations describing the context for that prediction; it is thus straightforward to use LM encoders to initialize POMDP policies (Chen et al., 2021; Janner et al., 2021). Our experiments use pre-trained GPT-2 (Radford et al., 2019) to initialize an interactive policy $\pi_{\text{LM}}$, which we fine-tune via imitation learning to perform interactive tasks. Different from the pure linguistic tasks that predict next words given the input sentence, we instead utilize the language models as policy by predicting the probability of the next action $a_t$, *i.e.* $a_t = \pi_{\text{LM}}(g, h_t, o_t)$. We note that such a formulation is broadly applicable to a variety of interactive tasks.

## 4 VirtualHome for Interactive Behavior Imitation

We consider embodied environments with 3D realistic scenes as the platform to test policies initialized from the pre-trained language model, as their goals and actions can be naturally represented as language, such as "put two plates on the kitchen table" and "grab apple". We further consider VirtualHome (Puig et al., 2018; 2020) for our experiments as we can easily get its symbolic representations that can be further serialized as English sentences. We note that the proposed approach is broadly applicable to other embodied environments with 3D realistic scenes, such as ALFRED (Shridhar et al., 2020) and iGibson (Shen et al., 2020).

VirtualHome is a 3D interactive environment to simulate household activities. Agents are represented as humanoid avatars. To investigate the contribution of language model pre-training on interactive behavior, we create VirtualHome-Imitation Learning, an improved version of VirtualHome with more challenging tasks and more diverse scenes and objects. Formally, the goal of an agent is to finish some household activities which is defined by a set of goal predicates. The agent takes an action $a_t$ based on the partial observation $o_t$, goal $g$, and history $h_t$ at each step. Next, we will describe the goals, observations, and actions provided in VirtualHome-Imitation Learning.

**Goal Space.** For each task, we define the goal as a set of predicates and multiplicities. For example, `inside(apple, fridge):2; inside(pancake, fridge):1;` means "put two apples and one pancake inside the fridge". In each task, the initial environment (including initial object locations), the goal predicates, and their orders and multiplicities are randomly sampled (see Appendix Section D). At training time, these environment configurations and goals reflect common-sense environment layouts (*e.g.* moving food from the refrigerator to the table). To further test the zero-shot generalization ability of different models, several test splits (discussed more in Section 5.1) feature other, less natural goals *e.g.* putting milk inside the dishwasher. In total, there are 59 different types of predicates. Each predicate can appear in a goal with a multiplicity between 0 and 3 (see Appendix D).

**Figure 2:** VirtualHome is a 3D interactive environment simulating household activities. It provides a symbolic graph representation of the partial observation. The action space changes over time based on the observation.

**Observation Space.** VirtualHome provides both graph-based and visual observations as shown in Figure 2; our experiments focused on graph-based observations. These represent each agent state as a scene graph, with nodes representing objects and edges describing their spatial relationships. The scene graph only contains object nodes appearing in the current partial observation.

**Action Space.** Agents can navigate in the environment and interact with objects. To interact with an object, the agent must predict an action name and the index of the interested object, *e.g.* open (5) to opening the object with index (5). The agent can only interact with objects that are in the current observation or execute the navigation actions, such as walk(bathroom). For some actions, such as open, the agent must be close to the object. There are also strict preconditions for actions, *e.g.* the agent must grab an object before it can put the object on a target position. As a result of these constraints, the subset of actions available to the agent changes at every timestep.

Different from text-based games (Narasimhan et al., 2015; Yao et al., 2020) that involve text as both the input and the output, VirtualHome is a 3D realistic environment, involving structured, graph-based observations and discrete, factorial actions. Moreover, our Experiment 2B (Section 7), shows that it is possible to learn how to interface between these observations / actions and an LM without any explicit string-based representation, a result that has no analog in any past work on text-based games. The tasks in VirtualHome are also challenging as they have lots of physical constraints and strict preconditions between actions. For example, the agent must grab the apple first before putting it on the kitchen table and must be close to the kitchen table. Text adventure environments in this sense are much easier as they have less physical constraints and less strict preconditions among actions compared with 3D environments. The objects' 3D information is also important for decision making in VirtualHome while text adventure environments do not have such 3D information. To train the models, we collect a set of expert trajectories in VirtualHome using regression planning (Korf, 1987). See Appendix Section C for more details.

# 5 EXPERIMENT SETUP

## 5.1 EVALUATION METRICS

We generate test sets that evaluate policies' ability to generalize in four ways: (1) generalization to familiar goals in novel (but in-distribution) environments; (2) generalization to novel goals; (3) generalization to abnormal initial environment states (defined by abnormal initial object locations); and (4) simultaneous generalization to both novel goals and abnormal initial states. We thus build four test subsets to evaluate a model from the four aspects. (See Appendix D for more details.)

**Normal.** In the "Normal" testing subset, the types of predicates and their counts are randomly sampled based on the same distribution as the training data. There are $2 \sim 10$ predicates in each task. The objects are initially placed in the environment according to common-sense layouts; (*e.g.* plates appear inside the kitchen cabinets rather than the bathtub). Note that even though goal predicates are drawn from the same distribution as the training data, the initial environments in the test set are different from the training set.

**Abnormal Initialization.** The objects are placed in random positions in the initial environment without common-sense constraints (*e.g.* apples may appear inside the dishwasher). There are $2 \sim 10$ goal predicates in each task.

**Zero-shot Combination.** The components of all goal predicates are never seen together during training (*e.g.* both plates and fridge appear in training goals, in(plate, fridge) appears only in the test set. There are $2 \sim 8$ goal predicates in each task.

**Abnormal Initialization + Zero-shot Combination.** Objects are initialized as in Abnormal Initialization above, and goals drawn from the same distribution as Zero-shot Combination above. There are $2 \sim 8$ goal predicates in each task.

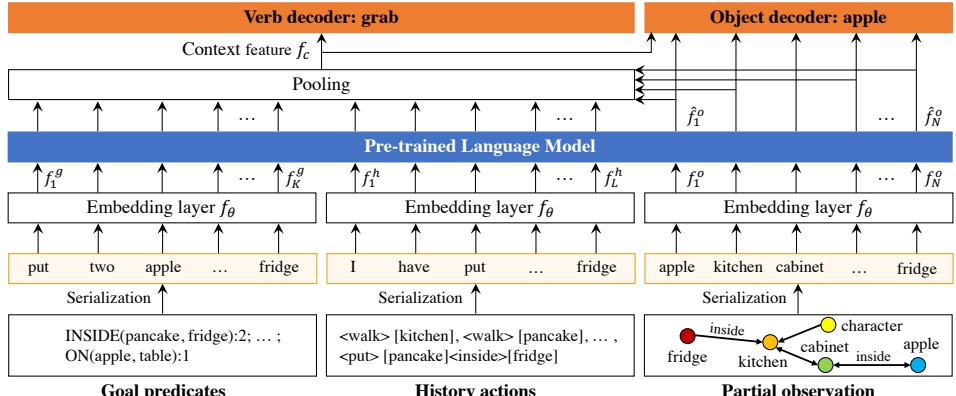

Figure 3: **Pre-trained language models for interactive imitation learning.** The objects in the current observation, the goal predicates, and history actions are first serialized as templated English phrases. We extract the tokens and their features from the phrases and send them to the pre-trained language model. The output of pre-trained language model are summarized into a context feature vector by average pooling which is then used for verb and object prediction.

For each test subset, we evaluate the success rate of different methods. A given episode is scored as successful if the policy completes its entire goal within 70 actions, where 70 is the max steps of the collected trajectories. For each model variant described below, we report the results of 10 training runs using 10 different random seeds. On each test subset, we use 5 different random seeds and test 100 tasks under each seed. Thus there are 5000 examples used for evaluating each model.

## 5.2 MODELS

To study the benefits of pre-training on the production of embodied interactive behaviors, we compare LM-based fine-tuning to a variety of baselines (see Appendix Section E for more details):

**LM (ft)** is the main model: a pre-trained GPT-2 transformer language model fine-tuned for a policy learning task (conditioned on goals, observations, and histories) as described in Section 3.3.

**LM (scratch)** uses the same model architecture and inputs, but is randomly initialized.

**LM (ft)** and **LM (scratch) w/o Hist** are ablations that do not condition on history information.

**MLP-N** and **MLP-1** take the goal, history actions, and the current observation as input and send them to the multilayer perceptron neural network (MLP) to predict actions. "MLP-1" has three more average-pooling layers than "MLP-N" that average the features of tokens in the goal, history actions, and the current observation, respectively, before sending them to the MLP layer.

**LSTM** uses a long short-term memory network (Hochreiter & Schmidhuber, 1997) to encode the history information. The hidden representation from the last time step together with the goal, and the current observation are used to predict the next action.

## 6 EXPERIMENT 1: LANGUAGE-BASED STATE AND ACTION ENCODINGS

Our first experiment aims to answer whether LMs can be used to initialize policies if state and action information is presented to them in a format that looks, to the greatest extent possible, like a standard language modeling problem. To do so, we encode the inputs to the policy—including observations, goals, and action histories—as templated English phrases (described in more detail below). The model architecture is shown in Figure 3. These phrases are passed directly to the LM (using its pre-trained word embeddings) and used to obtain contextualized token representations. These token representations are averaged, and used as input to action and object classifiers.

### 6.1 INPUT ENCODING

Inspired by the performance of recent large-language models, we choose to convert the observations, goal predicates, and history information, into text to take advantage of the pre-trained language model. To do this, we serialize the goal predicates, history actions, and observations as text and send them to the language model to learn the policy.

**Goal.** Each goal, consistsing of a sequence of predicates and multiplicities (*e.g.* "INSIDE(apple, fridge):2") is translated into a command in the format *put two apples inside the fridge*.

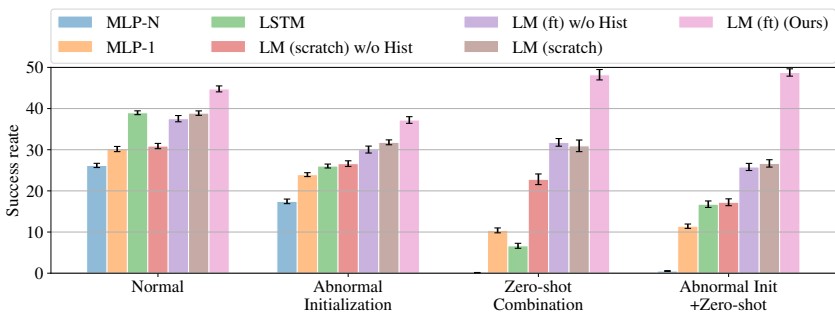

Figure 4: **Comparisons of the proposed method and baselines on different testing subsets.** "MLP-N", "MLP-1", and "LSTM" are baselines without using transformer (Vaswani et al., 2017). "LM (scratch) w/o Hist" and "LM (ft) w/o Hist" are based on the transformer architecture but do not use history in the input for decision making. "LM (scratch)" and "LM (ft) (Ours)" are based transformer and uses history in the input. The "scratch" means the transformer is trained from scratch on our data while "ft" means the transformer is pre-trained on language tasks and then fine-tuned on our data. Each experiment is performed 25 times with different random seeds. The averaged results are reported.

**Observation.** To encode the agent's partial observation, all the baselines and the proposed method use the symbolic graph representation instead of RGB visual representation as our focus is not to tackle the embodied perception problem. We aim at investigating the effectiveness of LM pretraining as a scaffold for learning control policies, and we thus utilize a common graph representation from previous works, *e.g.* (Puig et al., 2020) and (Liao et al., 2019). The graphs observations in VirtualHome consists of objects nodes in the current partial observation and the spatial relations among objects. Each object node has a name, *e.g.* "oven", a state description, *e.g.* "open, clean", and 3D world coordinates. Serialization of these scene graphs is the only part of the model that does not directly pass a string to the LM encoder. Instead, structured representations of each object are built up by combining name, attribute, and position embeddings; see Appendix A.1 for details. Solving the long-horizon tasks in VirtualHome is challenging because of the partial observability and strict preconditions among actions.

**History information.** Action histories are also converted into templated English sentences (*e.g.* "I have put the plate on the kitchen table and the apple inside the fridge") and passed directly to the language model.

## 6.2 OUTPUT DECODING

VirtualHome features a combinatorial action space involving verbs and objects. To accommodate this large action space, we factorize the action prediction, selecting verbs and objects separately.

**Verb prediction.** Given a vocabulary of verbs $\mathcal{V}$, we need to select one verb from them based on the current observation $o$, goal $g$, and history $h$. We take the output of the pre-trained language model and summarize them into a context feature $f_c$ by average pooling as shown in Figure 3. The context feature is then passed through a fully connected layer and a softmax layer to predict the probability $p(v_i|g,h,o)$ of selecting a verb $v_i$ at a single time step, where $v_i \in \mathcal{V}$.

**Object prediction.** Because of partial observability, the observed objects change over time. To enable the agent to only interact with objects in the current observation, we use an attention module to assign an attention score to each object and select the object with the highest attention score to interact with. We denote the feature of each object node after the pre-trained language model as $\hat{f}_i^o$. The attention score of each object node is represented as the inner product between its feature and the context feature: $s_i = f_c^T \hat{f}_i^o$. The attention scores $\{s_1, \cdots, s_N\}$ are then pass through a softmax layer to generate the probabilities $p(o_i|g,h,o)$ (where $i \in \{1, \cdots, N\}$) of the $N$ objects in the current observation $o$.

To maximize the likelihood of expert actions, we train the policy by jointly optimizing the verb and object predictions using the cross-entropy loss:

$$\mathcal{L} = \sum_{v_i} y_{v_i} \log(p(v_i|g,h,o)) + \sum_{o_i} y_{o_i} \log(p(o_i|g,h,o)), \tag{3}$$

where $y_{v_i}$ and $y_{o_i}$ are the binary labels of verb $v_i$ and object $o_i$, respectively.

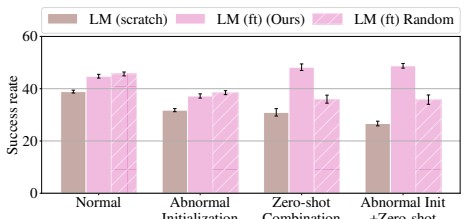

Figure 5: **Comparisons of pre-trained language models using natural and unnatural strings.** "LM (scratch)" and "LM (ft)" are the pre-trained language model using natural strings as input while "LM (ft) (Random)" uses random strings as input. In "LM (scratch)", the language model is trained from scratch on the collected data while "LM (ft)" and "LM (ft) (random)" fine-tune the pre-trained language model on the collected data.

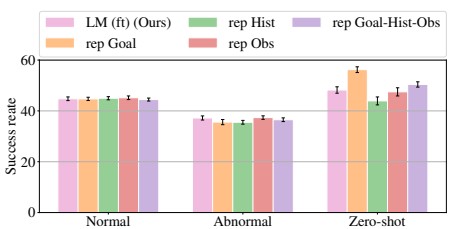

Figure 6: **Comparisons of pre-trained language encodings and learned encodings.** "LM (ft) (Ours)" uses the pre-trained language encodings. In "rep Goal", we use the learned encoding for goal and the pre-trained language encodings for history and observation. Similarly, "rep Hist" and "rep Obs" use the the learned encoding for history and observation, respectively. "rep Goal-Hist-Obs" uses the learned encoding for goal, history, and observation.

### 6.3 RESULTS

The results of different models on four test subsets are shown in Figure 4. In the **Normal** setting, where the test tasks are drawn from the same distribution as training tasks, policies initialized with a pre-trained language model, "LM (ft)", match the success rate of policies trained from scratch, "LM (scratch)". A similar trend is observed in the **Abnormal Initialization** setting. However, in the **Zero-shot Combination** and **Abnormal Init+Zero-shot** settings, where the test tasks requires generalization to novel goals or environment states that are never seen during training, we find that pre-trained policies, "LM (ft)", dramatically improve upon random initialization, "LM (scratch)", and all other baselines. The fact that "LM (ft)" performs slightly better on "Zero-shot Combination" and "Abnormal Init+Zero-shot" ($2 \sim 8$ predicates) than on "Normal" ($2 \sim 10$ predicates) is attributable to the fact that these sets involve fewer subgoals on average per task. Baselines without pre-training do not generalize to these new subgoal sets and perform worse on the zero-shot setting.

## 7 EXPERIMENT 2: NON-LINGUISTIC STATE AND ACTION ENCODINGS

We are interested in different ways of building interfaces between inputs and LMs, including string-based serialization and learned embedding. Experiment 1 demonstrated that language-based environment encodings contributed to effective generalization in LM-pretrained policies. Our two final experiments explore the sensitivity of this approach to the details of the input encoding.

### 7.1 EXPERIMENT 2A: NATURAL STRINGS V.S. UNNATURAL STRINGS

First, when using the pre-trained models' own string encoding mechanism, how important is it that strings passed as input resemble the training data? To evaluate this question, we replace the "natural language" tokens (e.g. serializing the goal "`ON(fork, table):1`" as *put one fork on the table*) with random ones (e.g. serializing `ON(fork, table)` as *brought wise character trees fine yet*). This is done by randomly permuting the entire vocabulary, mapping each token to a new index. The model architecture is the same as "LM (ft)" and the whole model is fine-tuned on our expert data. The results of permuting the vocabulary used to represent observation, goals, and history is in Figure 5.

In the "Normal" test condition, vocabulary scrambling has little effect: both natural and unnatural strings after fine-tuning are able to fit the testing tasks that are drawn from the same distribution as the training tasks. However, using unnatural strings significantly hurts the performance of the zero-shot setting where the testing tasks are unseen during training: even when not essential for in-distribution performance, natural string encodings are necessary to harness pretraining for stronger forms of generalization.

### 7.2 EXPERIMENT 2B: PRE-TRAINED LANGUAGE ENCODINGS V.S. LEARNED ENCODINGS

Given a non-linguistic task, if an effective string-based encoding cannot be generated arbitrarily, can such an encoding at least be *learned*? To answer this question, we retain the discrete, serial format of the goal, history, and observation representation, but replace the embedding layer from the pre-trained language model with a new embedding layer trained from scratch. We design four variants, some of which retain parameters from the original pre-trained LM: **rep Goal** uses a new

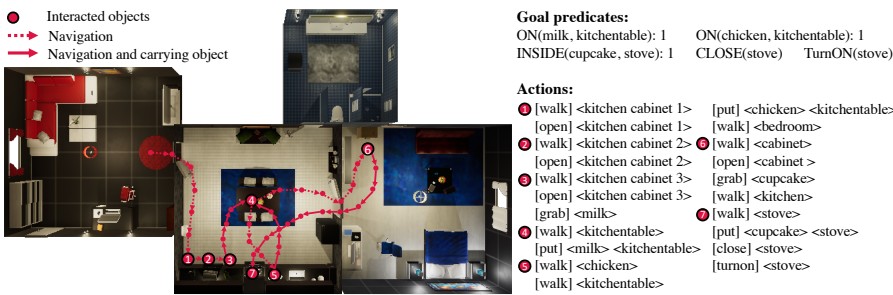

Figure 7: **Predicted trajectory and actions for a given household task.** The policy learned by fine-tuning the pre-trained language model successfully finishes the task described in the goal predicates. We highlight the key actions in the map, where the agent is finding, grabbing, or placing objects in the target positions.

embedding layer to encode goals while using the pre-trained embedding layer to encode observations and action histories. **rep Hist** and **rep Obs** use new learned embedding layer only for action histories and observations respectively. Finally **rep Goal-Hist-Obs** replaces the entire pre-trained language embedding layer, using a representation of the input to the policy learned entirely from scratch. Both our proposed model and the variants use serialized inputs. However, the variant models do not assume string-based inputs. The variant models use the embedding layer trained from scratch to represent each element in the serialized inputs, while our proposed model uses the language tokens generated by the tokenizer from the pre-trained language model and embeds them using the embedding layer from the pre-trained language model. See Section A.2 for more details. The results of these four variants and "ML (ft)" are shown in Figure 6. We find that replacing the embedding layer from the pre-trained language model has only a negligible impact on the performance compared with "ML (ft)". We thus conclude that the effectiveness of pre-training is not limited to string-based representations, but can be accessed in input representations learned entirely from scratch. To the best of our knowledge, neither template-string-based representations nor learned embeddings, combined with LM pretraining, has been applied to embodied decision-making problems in this way before; our main claim is that both approaches work better than learning from scratch in Virtual-Home given non-visual inputs.

## 8 QUALITATIVE RESULTS

In Figure 7, we show one example of "LM (ft)" completing the household organization task in VirtualHome. Given a goal described by a set of goal predicates,"ON(milk, kitchentable):1; ON(chicken, kitchentable):1; INSIDE(cupcake, stove):1; CLOSE(stove); TurnON(stove)". The agent first explores the environment until it finds the milk. Then the agent grabs the milk, walks to the kitchen table, and puts the milk on the kitchen table. The agent moves to the next subtask until it finishes all the subtasks. Completing all the subtasks is challenging because of the partial observability, large action spaces, and long time horizons. Furthermore, there are strict preconditions between actions, *e.g.* to put the milk on the kitchen table, the agent must grab the milk first. The proposed method that serializes observations, goals, and history information as English strings and takes advantage of the pre-trained language model is able to finish the task efficiently.

## 9 CONCLUSION

We have presented a set of experiments studying the effect of pre-training on embodied interactive behaviors in VirtualHome. To leverage LM pre-training, we described a policy representation in which observations, goals, and history information were serialized as templated English strings, and found that language models fine-tuned as policies with these inputs enjoyed substantial benefits in zero-shot task and environment generalization over randomly initialized baselines. Additional experiments exploring the sensitivity of pre-training to the encoding of the policy-learning task showed that poorly designed string encodings removed these generalization benefits, but that effective encoding layers could be learned from scratch in the absence of a string-based goal and observation representation. These results suggest that language model pre-training produces representations not just of language, but of the abstract structure of goals and plans, and that these representations might be useful in a variety of tasks beyond language processing.

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

# Appendix

In this appendix, we first provide more implementation details of the proposed model in Section A. We then show the additional results in Section B, including the analysis of using history information as input and the analysis of attention weights from language models. In Section C, we give more details of collecting the expert data. Section D lists the goal predicates in different test subsets. The implementation details of baselines are shown in Section E.

## A    MORE IMPLEMENTATION DETAILS OF THE PROPOSED MODEL

In Section A.1, we provide more details of the model architecture used in the main paper Section 6. We then introduce the training detail in Section A.3.

### A.1    MODEL ARCHITECTURE DETAILS

In the main paper Section 6, we introduced the model architecture used for training an interactive policy. Our model consists of three parts, *i.e.* inputs, the pre-trained language model, and outputs. We take the goal $g$, history $h_t$, and the current partial observation $o_t$ as inputs and send them to the policy network initialized with the pre-trained language model. The output action $a_t$ consists of a verb and an object. For brevity, we will omit the time subscript $t$ from now on.

In VirtualHome, the partial observation $o$ of the environment state can be represented as a scene graph, with nodes representing objects and edges describing their spatial relationships. Each object node has a name, *e.g.* "oven", a state description, *e.g.* "open, clean", and world coordinates. In the main paper Section 6, we briefly describe how to encode the observation. In this part, we provide more details of encoding the name, state, and position of each object in the current observation. Figure A1 shows the model architecture we used to encode the observation.

Instead of flattening the graph observation as plain text, we keep the graph entities in observations. This is because graphs representing agent observations are very large, involve a large number of distinct relations and LMs are not effective at numerical reasoning, such as 3D position information. For example, we have to use many sentences to describe the relations between objects, *e.g.* "the apple is on the kitchen table, the apple is close to the banana, the kitchen table is close to the fridge ...". The pre-trained GPT2 model from Hugging Face (Wolf et al., 2019) can feed in a maximum of 1024 inputs. Describing every object and the relations between objects are too long to be encoded with the LMs. In addition, the numerical features, such as 3D position information, maybe could be translated into a naturalistic sentence-like format by saying "the apple is in position ($x = 0.02, y = 0.07, z = 0.01$) and is $0.5$ meters away from the banana ... ". but cannot be encoded efficiently using language models as LMs are not effective at numerical reasoning. We thus utilize a graph encoder to remain as close as possible to a natural language input without producing string-based graph representations that are too long to encode with the LM.

**Name encoding.** For each object node, we serialize its object name as an English phrase $s^o$. For each word $w_i^o$ in the English phrase $s^o$, we extract its token $t_i^o$ and features using the tokenizer and the embedding layer of the pre-trained language model, respectively. Since one object name might generate several English tokens using the tokenizer from the pre-trained language model, *e.g.* the tokens of "kitchencabinet" is $[15813, 6607, 66, 6014, 316]$, we take the averaged features of all the tokens in the object name and obtain a "name" feature $f_i^{o,\text{name}}$ for each object node as shown in Figure A1.

**State encoding.** Given the current observation $o$, some objects have a state description, *e.g.* "oven: open, clean". There are six types of object states in the whole training dataset, including, "clean", "closed", "off", "on", "open", and "none". Thus for each object node, we use a 6-dim vector to represent its state. Taking the "oven" as an example, if the oven is open and clean, its state vector would be $[1, 0, 0, 0, 1, 0]$. This state vector is then passed through a fully-connected layer to generate a state feature $f_i^{o,\text{state}}$ of object $o_i$.

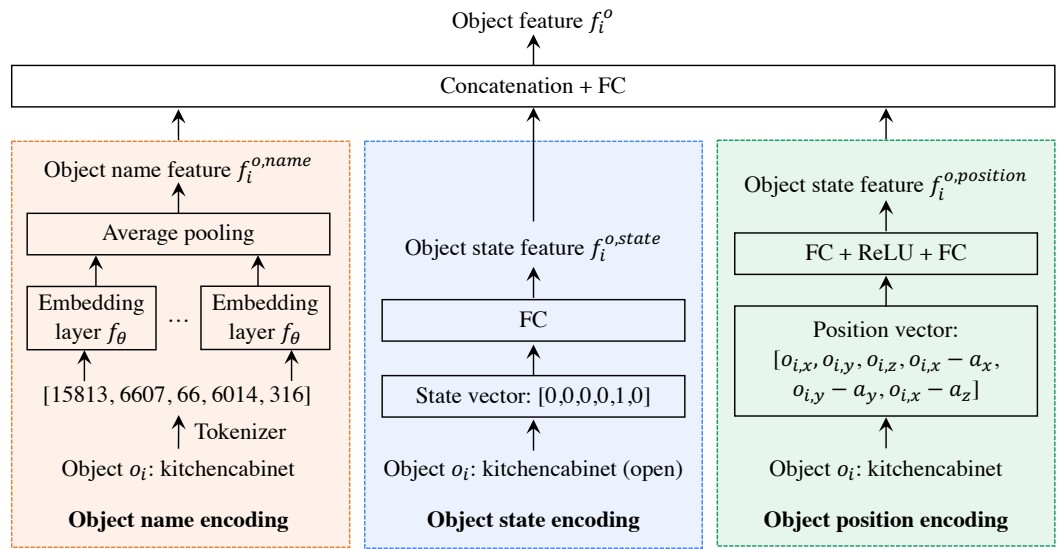

Figure A1: **Object encoding.** In VirtualHome, the partial observation $o$ of the environment state can be represented as a scene graph, with nodes representing objects and edges describing their spatial relationships. Each object node in the observation has a name, a state description, and world coordinates. **Object name encoding:** for each object node, we serialize its object name as an English phrase. For each word in the English phrase, we extract its tokens and features using the tokenizer and the embedding layer of the pre-trained language model, respectively. We take the averaged features of all the English tokens in the object name and obtain a "name" feature $f_i^{o,\text{name}}$ for each object node. **Object state encoding:** there are six types of object states in the whole training dataset, including, "clean", "closed", "off", "on", "open", and "none". Thus for each object node, we use a 6-dim vector to represent its state. This state vector is then passed through a fully-connected layer to generate a state feature $f_i^{o,\text{state}}$ of object $o_i$. **Object position encoding:** we concatenate the world coordinates $\{o_{i,x}, o_{i,y}, o_{i,z}\}$ of each object and their spatial distance to the agent $\{a_x, a_y, a_z\}$ to generate a position vector $[o_{i,x}, o_{i,y}, o_{i,z}, o_{i,x} - a_x, o_{i,y} - a_y, o_{i,z} - a_z]$. This position vector is then passed through two fully-connected layers with a ReLU layer in the middle to generate a position feature $f_i^{o,\text{position}}$ of object $o_i$. The final feature $f_i^o$ of each object node is obtained by passing the concatenation of its name feature $f_i^{o,\text{name}}$, state feature $f_i^{o,\text{state}}$, and position feature $f_i^{o,\text{position}}$ through a fully-connected layer.

**Position encoding.** To encode the position information of each object $o_i$, we take their world coordinates $\{o_{i,x}, o_{i,y}, o_{i,z}\}$ and their spatial distance to the agent $\{a_x, a_y, a_z\}$ to generate a position vector $[o_{i,x}, o_{i,y}, o_{i,z}, o_{i,x} - a_x, o_{i,y} - a_y, o_{i,z} - a_z]$. This position vector is then passed through two fully-connected layers with a ReLU layer in the middle to generate a position feature $f_i^{o,\text{position}}$ of object $o_i$.

The final feature $f_i^o$ of each object node is obtained by passing the concatenation of its name feature $f_i^{o,\text{name}}$, state feature $f_i^{o,\text{state}}$, and position feature $f_i^{o,\text{position}}$ through a fully-connect layer. The observation at a single step can be written as a set of features $\{f_1^o, \cdots, f_N^o\}$, where $N$ is the number of objects in the current observation $o$.

## A.2 MODEL DETAILS IN EXPERIMENT 2B

In the main paper Section 7.2, we introduced four model variants that replace the embedding layer from the pre-trained language model with a new embedding layer trained from scratch. In this section, we provide more details of the proposed model and the four variants. We show the model architectures of the proposed model, "rep Goal", "rep Hist", "rep Obs", and "rep Goal-Hist-Obs" in Figure A2, Figure A3, Figure A4, Figure A5, and Figure A6, respectively.

To train the variant models in Experiment 2B, we assign an index ID to each "element" in the dataset. The "element" could be: 1) the object names in the observation, e.g. "fridge", "kitchentable", and other objects if the agent is in the kitchen; 2) the object names in the goal predicates, e.g. "plate", "kitchentable", and "fridge" in the goal predicate "ON(plate, kitchentable); INSIDE(plate, fridge)"; or 3) the actions, e.g. "walk" and "bathroom" in the action "[walk] < bathroom >". We thus ob-

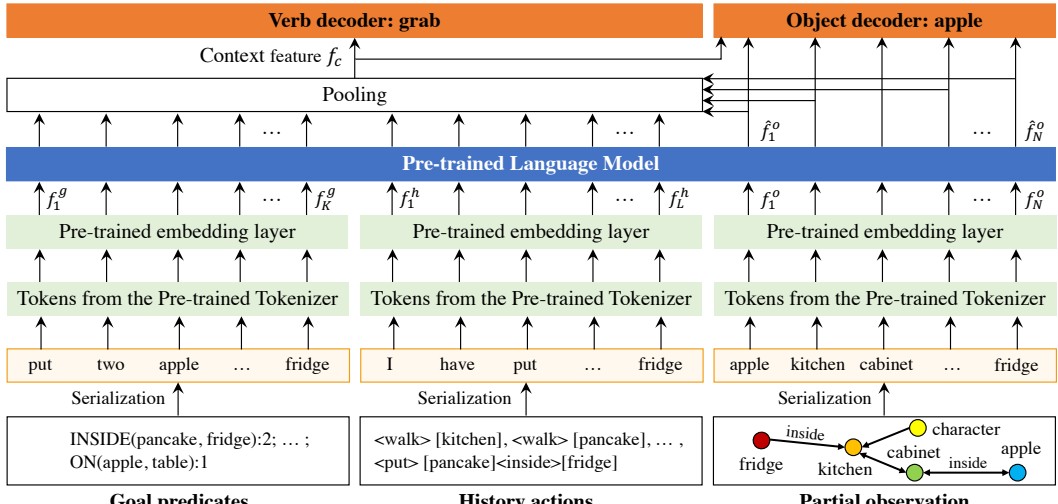

Figure A2: **Model architecture of the propose model in the main paper Section 6**.

tain a set of "elements" from the whole dataset, such as ["fridge", "kitchentable", "plate", "walk", "bathroom", ..., "none"]. We then assign an index ID (starting from 0) to each of them, such as { "fridge": 0, "kitchentable": 1, "plate": 2, "walk": 3, "bathroom": 4, ..., "none": MAX ID }. Then we use a one-hot vector to represent each "element" based on their index ID. For example, "fridge" could be represented as a vector of [1, 0, 0, 0, ...].

To train the variant models in Experiment 2B, taking the "rep Goal" in Figure A3 as an example, we first use this one-hot vector to represent all the "elements" in the goal predicates. The one-hot vector is then sent to a learned embedding layer that is trained from scratch to generate a new feature representation for each "element" in the goal predicates $[f_1^{g(\text{learned})}, f_2^{g(\text{learned})}, \ldots, f_K^{g(\text{learned})}]$. To encode the partial observation and history actions, we keep the same encoding as the proposed model in Figure A2, where we first use the tokenizer from the pre-trained language model to generate the language token for each "word" in the partial observation and history actions and then send the tokens to the embedding layers from the pre-trained language model to generate a feature representation for each "word", *i.e.* $[f_1^{o(\text{pretrained})}, f_2^{o(\text{pretrained})}, \ldots, f_N^{o(\text{pretrained})}]$ for the partial observation and $[f_1^{h(\text{pretrained})}, f_2^{h(\text{pretrained})}, \ldots, f_L^{h(\text{pretrained})}]$ for the history actions. The output features of goal predicates, partial observation, and history actions are then concatenated and sent to the pre-trained language model for training.

Similarly, in the variant model "rep Hist", the history actions use the learned embeddings while the goal predicates and partial observation use the pre-trained embeddings. In "rep Obs" and "rep Goal-Hist-Obs", we use learned embeddings as described above to encode the "name" information of each object in the current observation. We use the same object state encoding and object position encoding described in Section A.1 to encode the "state" and "position" information of each object.

Note that the representations of "elements" and "words" are different. For example, "kitchentable" is a single "element" and has an index ID of "1" in the models with learned embeddings, but its English token extracted from the pre-trained GPT2 is "[15813, 2395, 429, 540]". Another example is "plate" having an index ID of "2" in the models with learned embeddings but its English token extracted from the pre-trained GPT2 is "[6816]".

## A.3 TRAINING DETAILS

Our proposed approach and baselines are trained on Tesla 32GB GPUs. We train every single model on 1 Tesla 32GB GPU. Each experiment was trained on 10 different seeds and tested on 5 different seeds. All experiments used the Adam optimizer with the learning rate $1e^{-4}$. We utilize a standard

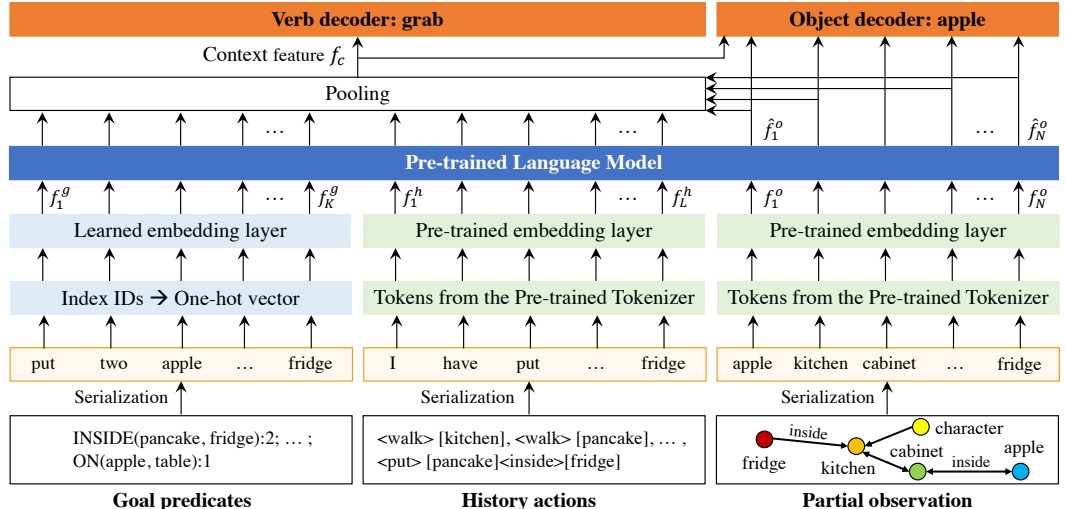

Figure A3: **Model architecture of "rep Goal" used in the main paper Section 7.2**.

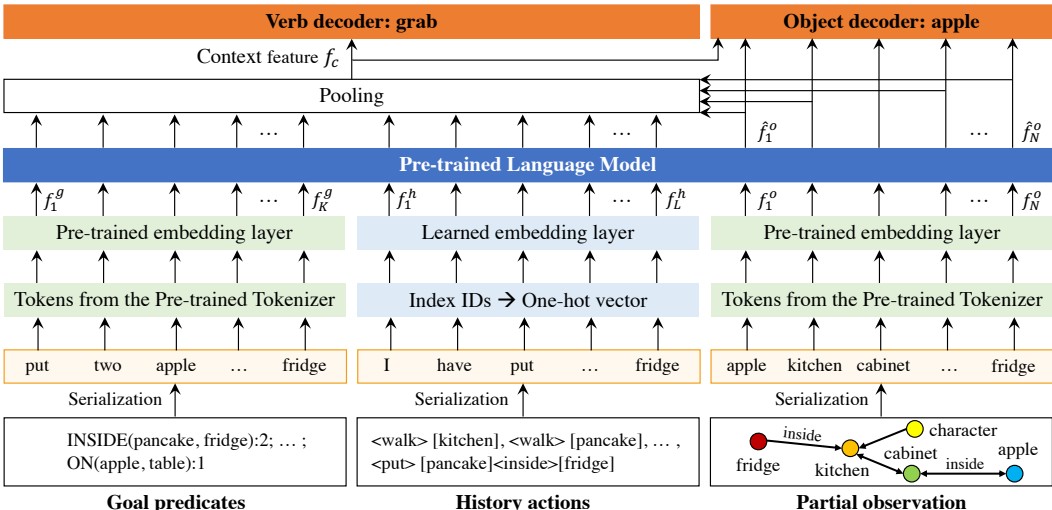

Figure A4: **Model architecture of "rep Hist" used in the main paper Section 7.2**.

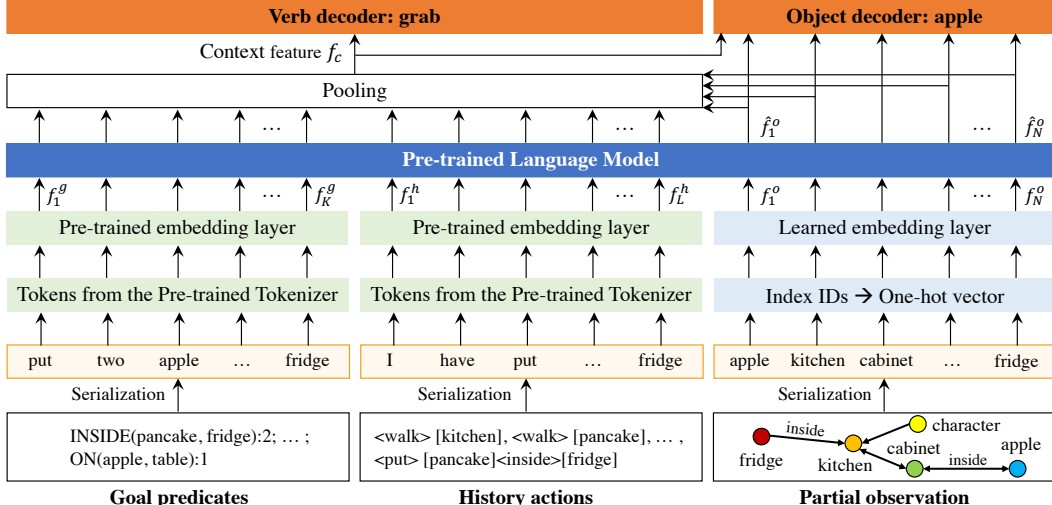

Figure A5: **Model architecture of "rep Obs" used in the main paper Section 7.2**.

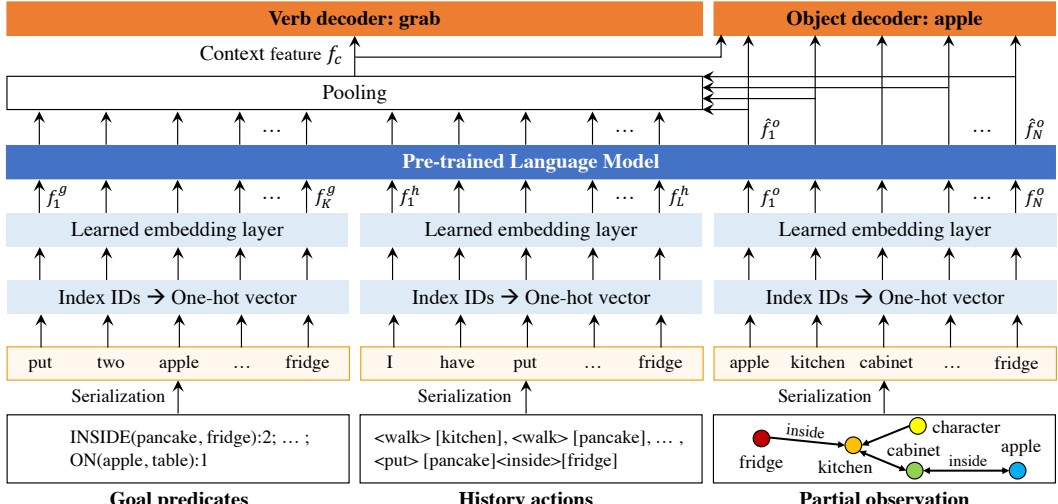

Figure A6: **Model architecture of "rep Goal-Hist-Obs" used in the main paper Section 7.2**.

pre-trained language model, GPT-2 that is trained on Webtext dataset (Radford et al., 2018), in our experiments by using Huggingface library (Wolf et al., 2019) .

# B   ADDITIONAL RESULTS

In the main paper Section 6, we show that the policy with pre-trained language model outperforms other baselines by a large margin on the zero-shot setting. In this appendix Section B.1, we further discuss the effectiveness of using history information. In Section B.2, we show the attention weights from the attention layers in the language model after fine-tuned on our data.

## B.1   SERIALIZE HISTORY INFORMATION AS SENTENCES

Given the partial observability of the tasks, the models require history information to know what subtasks have been finished. To take advantage of the pre-trained language model, we serialize history actions as sentences and send them into the language model for action prediction. As shown in the main paper Figure 4, such an operation brings an improvement on all four test subsets, *i.e.* "LM (ft)" outperforms "LM (ft) w/o Hist" for models with pre-training and "LM (scratch)" outperforms "LM (scratch) w/o Hist" for models without pre-training.

In the **Zero-shot Combination** and **Abnormal Init+Zero-shot** settings, we find that the improvement brought by the history sentences is more obvious for models with pre-training, *i.e.* the difference between "LM (ft)" and "LM (ft) w/o Hist" is much larger than the difference between "LM (scratch)" and "LM (scratch) w/o Hist".

In addition, we find that "LM (ft)" is slightly better than "LSTM" on the **Normal** setting, but "LM (ft)" outperforms "LSTM" by more than 40% on the **Zero-shot Combination** setting. Even "LSTM" takes extra history information from the hidden representation of previous steps, its performance drops dramatically on the zero-shot setting with unseen tasks.

Together, serializing history information as sentences is helpful for embodied interactive behaviors. Such an operation enables us to utilize the pre-trained language model and outperforms the LSTM model significantly. The advantage of using pre-training is more obvious on the **Zero-shot Combination** setting.

## B.2   ATTENTION WEIGHTS

In the proposed model "LM (ft)", we utilize the pre-trained GPT-2 (Radford et al., 2019) to model the input sentences. To better understand how does the policy with a pre-trained language model make decisions, we extract the attention weight from self-attention layers in GPT-2 (GPT-2 is a

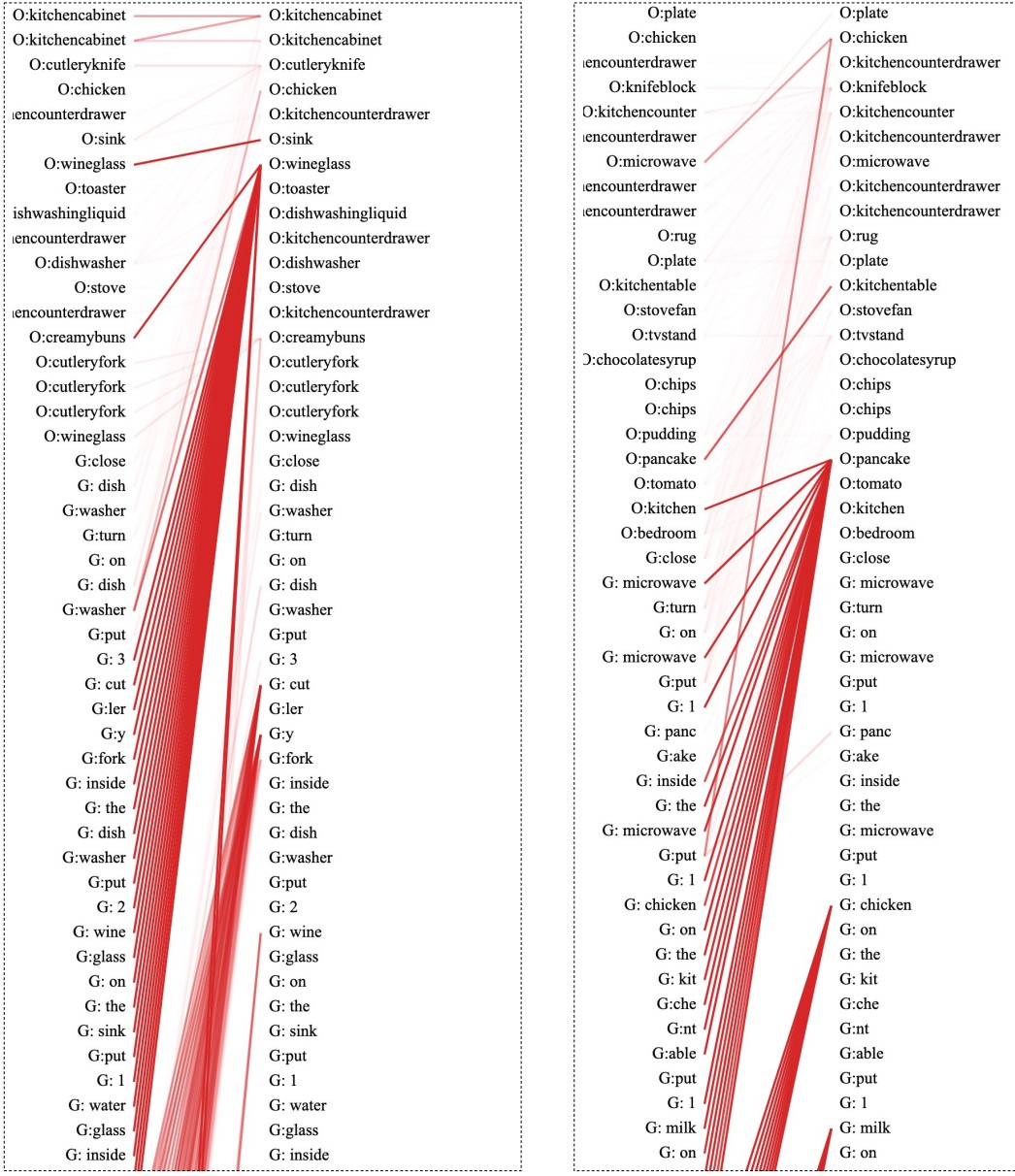

Figure A7: **Attention weights of a layer named "Head 3 Layer 2".** We show attention weights on two different tasks. We find that "Head 3 Layer 2" is able to capture objects in the goal predicates, such as "wineglass" and "cutleryfork" in the left figure, and "pancake" and "chicken" in the right figure (the figures are cropped for visualization).

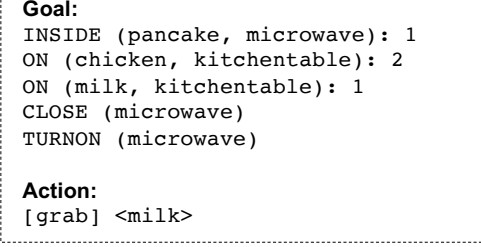

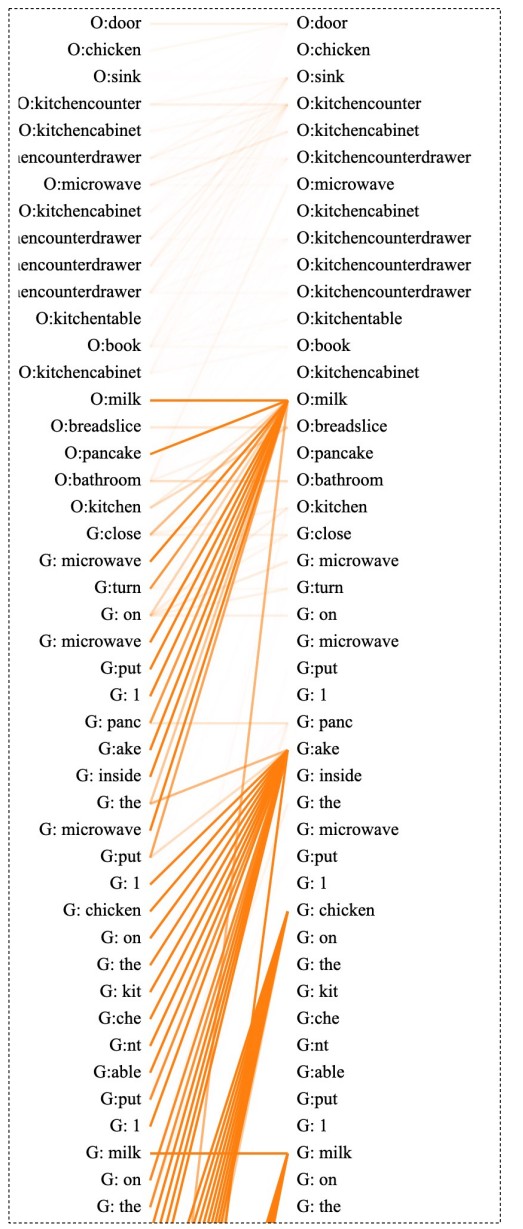
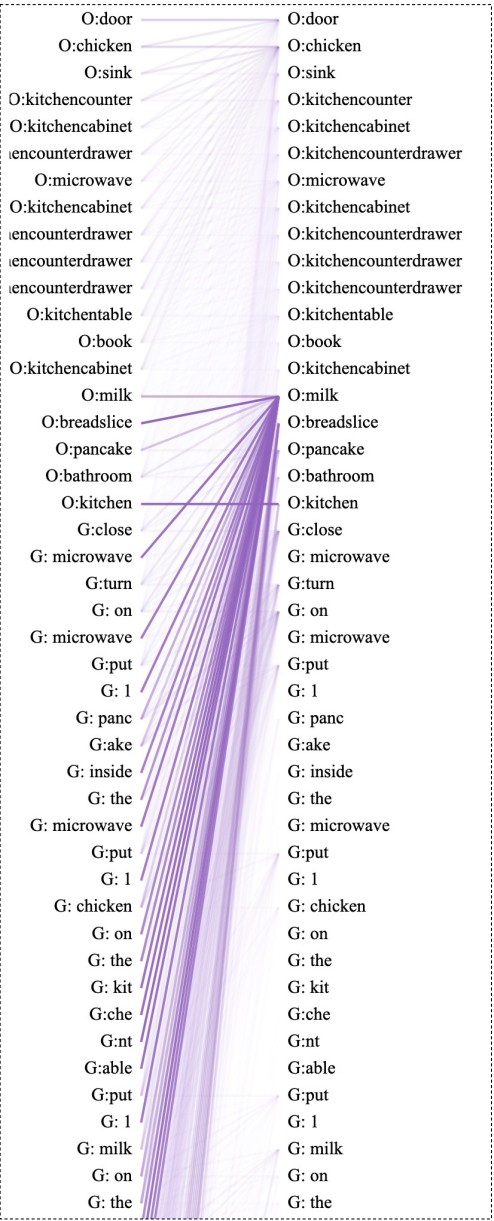

Figure A8: **Attention weights of layers named "Head 1 Layer 2" (left) and "Head 4 Layer 11" (right).** Given the goal predicates, history, and the current observation, "LM (ft)" predicts the next action, *i.e.* "grab milk". We find that "Head 1 Layer 2" is able to capture objects in the goal predicates, such as "milk", "pancake", and "chicken" while "Head 4 Layer 11" focuses on the interacted object in the predicted action, such as "milk".

Transformer model (Vaswani et al., 2017)). In Figure A7 and Figure A8, we show the attention weights from the input (left) to the output (right) of GPT-2. The order of tokens in the input and ouput is observation, goal, and history.

Figure A7 illustrates the attention weights of a layer named "Head 3 Layer 2". We show attention weights on two different tasks. We find that "Head 3 Layer 2" is able to capture objects in the goal predicates, such as "wineglass" and "cutleryfork" in the left figure, and "pancake" and "chicken" in the right figure (the figures are cropped for visualization).

Figure A8 illustrates the attention weights of layers named "Head 1 Layer 2" (left) and "Head 4 Layer 11" (right). Given the goal predicates, history, and the current observation, "LM (ft)" predicts the next action, *i.e.* "grab milk". We find that "Head 1 Layer 2" is able to capture objects in the goal predicates, such as "milk", "pancake", and "chicken" while "Head 4 Layer 11" focuses on the interacted object in the predicted action, such as "milk".

We find that the attention weights from different self-attention layers are significantly different, some self-attention layers assign high attention weight to objects in the goal predicates while some layers focus on the interacted object. There are also some layers that do not have interpretable meanings. The attention weights just provide us an intuition of how does the internal language model works, more quantified results are reported in the main paper Figure 4, Figure 5, and Figure 6.

## C EXPERT DATA COLLECTION

In this section, we provide more details of expert data collection described in the main paper Section 4. To train the models, we first collect a set of expert trajectories using regression planning (RP) (Korf, 1987). We follow the implementation of the regression planner in (Puig et al., 2020). Given a task described by goal predicates, the planner generates an action sequence to accomplish this task. As shown in Figure A9, the agent has a belief about the environment, *i.e.* an imagined distribution of object locations. As the agent explores the environment, its belief of the world becomes closer to the real world. At every step, the agent updates its belief based on the latest observation (see (Puig et al., 2020)), finds a new plan using the regression planner, and executes the first action of the plan. If the subtask (described by the goal predicate) has been finished, the agent will select a new unfinished subtask, otherwise, the agent will keep doing this subtask until finish it.

As in much of past work (Shridhar et al., 2020; Shen et al., 2020; Puig et al., 2020), the planner used to generate training data has access to privileged information, such as full observation of the environment and information about the pre-conditions and effects of each action, while the actual deployed policy do not have such information. The planner allows an agent to robustly perform tasks in partially observable environments and generate expert trajectories for training and evaluation.

Eventually, we generate $80,416$ trajectories for training and $3,758$ trajectories for validation. Each trajectory has a goal, an action sequence, and the corresponding observations after executing each action.

## D TEST SUBSETS

We introduced four subsets for interactive evaluation in the main paper Section 5. In Table A1, we provide a detailed description of each subset, including the count of goal predicate types and the number of goal predicates in each task. The **Normal** setting has 37 goal predicates in total and each task has $2 \sim 10$ goal predicates. The tasks are drawn from the same distribution as the training tasks. The **Abnormal Initialization** setting also has 37 goal predicates and each task has $2 \sim 10$ goal predicates. The objects are randomly placed in the initial environment. The **Zero-shot Combination** setting has 22 goal predicates in total and each task has $2 \sim 8$ goal predicates. The tasks are never seen during training. The **Abnormal Init+Zero-shot** setting also has 22 goal predicates and each task has $2 \sim 8$ goal predicates. The objects are randomly placed in the initial environment and the tasks are never seen during training.

In Figure A2 and Figure A3, we list the full goal predicates used in each subset. The goal predicates of a task are randomly sampled from the corresponding predicates pool.

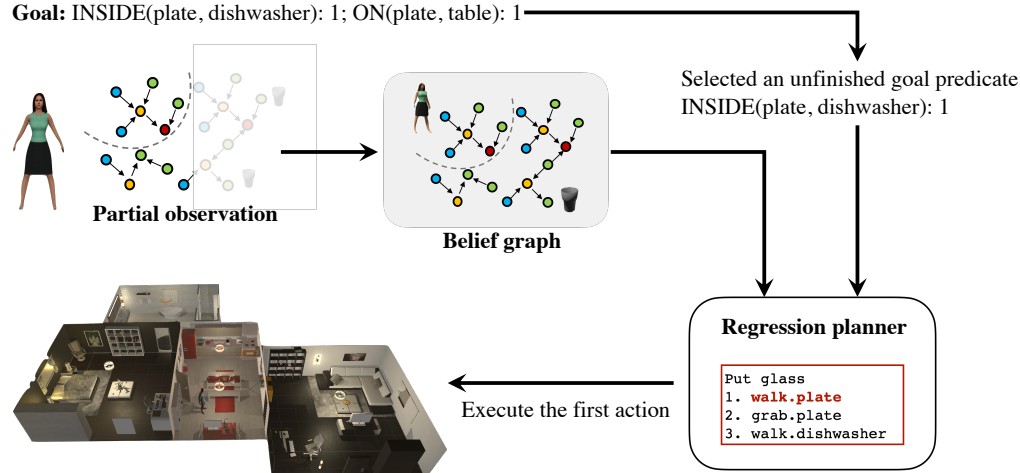

Figure A9: **Regression planner**. Given a task described by goal predicates, the planner generates an action sequence to accomplish this task. The agent has a belief about the environment, *i.e.* an imagined distribution of object locations. As the agent explores the environment, its belief of the world becomes closer to the real world. At every step, the agent updates its belief based on the latest observation (see (Puig et al., 2020)), finds a new plan using the regression planner, and executes the first action of the plan. If the subtask (described by the goal predicate) has been finished, the agent will select a new unfinished subtask, otherwise, the agent will keep doing this subtask until finish it.

Table A1: **Summary of four test subsets.** For each test subset, we show the count of goal predicate types and the number of goal predicates in each task.

| Test Sets | Pred. Types | #Pred. Per Task | Compared with the training set |
|---|---|---|---|
| **Normal test** | 37 | $2 \sim 10$ | Tasks are drawn from the same distribution as training tasks. |
| **Abnormal Initialization** | 37 | $2 \sim 10$ | The objects are randomly placed in the initial environment. |
| **Zero-shot Combination** | 22 | $2 \sim 8$ | Tasks are unseen during training. |
| **Abnormal Init+Zero-shot** | 22 | $2 \sim 8$ | The combination of Abnormal initialization and Zero-shot combination. |

## E   IMPLEMENTATION DETAILS OF BASELINES

We introduced six baselines in the main paper Section 5.2, including "MLP-N", "MLP-1", "LSTM", "LM (scratch) w/o Hist", "LM (ft) w/o Hist", and "LM (scratch)". "LM (scratch)" uses the same architecture as the proposed method "LM (ft)" in Figure 3 in the main paper. "LM (scratch) w/o Hist" and "LM (ft) w/o Hist" also have a similar architecture as "LM (ft)" except that they do not take the history information as input. Thus in this section, we provide more details of "MLP-N", "MLP-1", and "LSTM".

**MLP-N.** The model architecture of "MLP-N" is shown in Figure A10. The input and output are the same as "LM (ft)" as we introduced in the main paper Section 6 and this appendix Section A.1. The difference is that instead of sending the concatenated features of the goal, history, and observation to the pre-trained language model, "MLP-N" sends them to an MLP followed by an average pooling layer to generate the context feature $f_c$. The MLP consists of two fully-connected layers with a ReLU layer in the middle. The verb decoder and object decode are the same as "LM (ft)".

**MLP-1.** The model architecture of "MLP-1" is shown in Figure A11. The input and output are the same as "LM (ft)" as we introduced in the main paper Section 6 and this appendix Section A.1. "MLP-1" obtains an averaged feature $f_g$ of the task goal by averaging the features of all the tokens in the goal sentences $\{f_1^g, \cdots, f_K^g\}$. Similarly, "MLP-1" obtains an averaged feature of the history $f_h$ and an averaged feature of the observation $f_o$ by averaging the features of all the tokens in the history sentences $\{f_1^h, \cdots, f_L^h\}$ and all the observed objects $\{f_1^o, \cdots, f_N^o\}$, respectively. The averaged goal feature $f_g$, the averaged history feature $f_h$, and the averaged observation feature $f_o$ are concatenated and sent to an MLP to generate the context feature $f_c$. The MLP consists of two fully-connected

Table A2: **Goal predicates in the Normal and Abnormal Initialization settings.**

```
ON(cutleryfork, kitchentable):0 ∼ 3
ON(plate, kitchentable):0 ∼ 3
ON(waterglass, kitchentable):0 ∼ 3
ON(wineglass, kitchentable):0 ∼ 3
INSIDE(cutleryfork, dishwasher):0 ∼ 3
INSIDE(plate, dishwasher):0 ∼ 3
INSIDE(waterglass, dishwasher):0 ∼ 3
INSIDE(wineglass, dishwasher):0 ∼ 3
INSIDE(cutleryfork, sink):0 ∼ 3
INSIDE(plate, sink):0 ∼ 3
INSIDE(waterglass, sink):0 ∼ 3
INSIDE(wineglass, sink):0 ∼ 3
INSIDE(milk, fridge):0 ∼ 3
INSIDE(chicken, fridge):0 ∼ 3
INSIDE(cupcake, fridge):0 ∼ 3
INSIDE(pancake, fridge):0 ∼ 3
INSIDE(poundcake, fridge):0 ∼ 3
ON(milk, kitchentable):0 ∼ 3
ON(chicken, kitchentable):0 ∼ 3
ON(cupcake, kitchentable):0 ∼ 3
ON(pancake, kitchentable):0 ∼ 3
ON(poundcake, kitchentable):0 ∼ 3
INSIDE(chicken, microwave):0 ∼ 3
INSIDE(cupcake, microwave):0 ∼ 3
INSIDE(pancake, microwave):0 ∼ 3
INSIDE(poundcake, microwave):0 ∼ 3
INSIDE(chicken, stove):0 ∼ 3
INSIDE(cupcake, stove):0 ∼ 3
INSIDE(pancake, stove):0 ∼ 3
INSIDE(poundcake, stove):0 ∼ 3
CLOSE(stove):0 ∼ 1
CLOSE(dishwasher):0 ∼ 1
CLOSE(microwave):0 ∼ 1
CLOSE(fridge):0 ∼ 1
TurnON(stove):0 ∼ 1
TurnON(dishwasher):0 ∼ 1
TurnON(microwave):0 ∼ 1
```

Table A3: **Goal predicates in the Zero-shot Combination and Abnormal Initialization+Zero-shot Combination settings.**

```
INSIDE(milk, dishwasher):0 ∼ 3
INSIDE(chicken, dishwasher):0 ∼ 3
INSIDE(cupcake, dishwasher):0 ∼ 3
INSIDE(pancake, dishwasher):0 ∼ 3
ON(milk, sink):0 ∼ 3
ON(chicken, sink):0 ∼ 3
ON(cupcake, sink):0 ∼ 3
ON(pancake, sink):0 ∼ 3
INSIDE(cutleryfork, fridge):0 ∼ 3
INSIDE(plate, fridge):0 ∼ 3
INSIDE(waterglass, fridge):0 ∼ 3
INSIDE(wineglass, fridge):0 ∼ 3
INSIDE(cutleryfork, microwave):0 ∼ 3
INSIDE(plate, microwave):0 ∼ 3
INSIDE(waterglass, microwave):0 ∼ 3
INSIDE(wineglass, microwave):0 ∼ 3
INSIDE(milk, microwave):0 ∼ 3
INSIDE(cutleryfork, stove):0 ∼ 3
INSIDE(plate, stove):0 ∼ 3
INSIDE(waterglass, stove):0 ∼ 3
INSIDE(wineglass, stove):0 ∼ 3
INSIDE(milk, stove):0 ∼ 3
```

layers with a ReLU layer in the middle. The verb decoder takes the context feature $f_c$ as input and outputs a verb prediction. Different from "LM (ft)", the object decoder takes the object features right after the "Embedding layer $f_\theta$" box as input to predicate the interacted object.

**LSTM.** The model architecture of "LSTM" is shown in Figure A12. The input and output are the same as "LM (ft)" as we introduced in the main paper Section 6 and this appendix Section A.1. "LSTM" has a similar architecture as "MLP-N" in Figure A10. The difference is in the ways to compute the context feature $f_c$. The feature $\hat{f}_c$ after the pooling layer is first concatenated with the hidden feature $h_{t-1}$ from the last step and then sent to LSTM (Hochreiter & Schmidhuber, 1997). LSTM outputs the context feature $f_c$ and a hidden feature $h_t$ that will be used for the next step. The verb decoder and object decode are the same as "MLP-N".

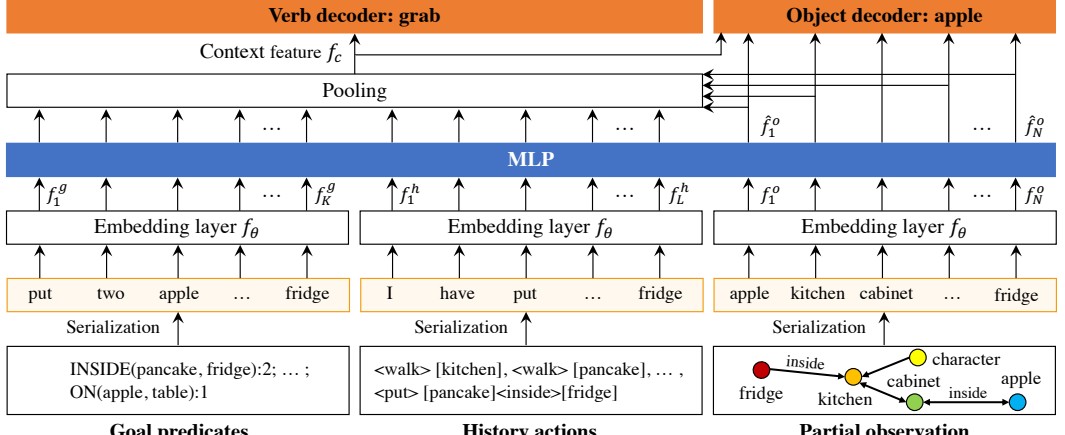

Figure A10: **Baseline MLP-N.** The input and output are the same as "LM (ft)" as we introduced in the main paper Section 6 and this appendix Section A.1. The difference is that instead of sending the concatenated features of the goal, history, and observation to the pre-trained language model, "MLP-N" sends them to an MLP followed by an average pooling layer to generate the context feature $f_c$. The MLP consists of two fully-connected layers with a ReLU layer in the middle. The verb decoder and object decode are the same as "LM (ft)".

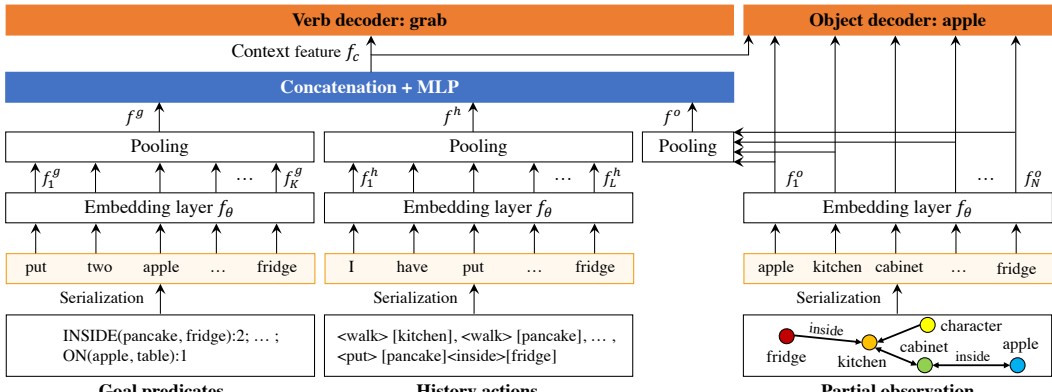

Figure A11: **Baseline MLP-1.** The input and output are the same as "LM (ft)" as we introduced in the main paper Section 6 and this appendix Section A.1. "MLP-1" obtains an averaged feature $f_g$ of the task goal by averaging the features of all the tokens in the goal sentences $\{f_1^g, \cdots, f_K^g\}$. Similarly, "MLP-1" obtains an averaged feature of the history $f_h$ and an averaged feature of the observation $f_o$ by averaging the features of all the tokens in the history sentences $\{f_1^h, \cdots, f_L^h\}$ and all the observed objects $\{f_1^o, \cdots, f_N^o\}$, respectively. The averaged goal feature $f_g$, the averaged history feature $f_h$, and the averaged observation feature $f_o$ are concatenated and sent to an MLP to generate the context feature $f_c$. The MLP consists of two fully-connected layers with a ReLU layer in the middle. The verb decoder takes the context feature $f_c$ as input and outputs a verb prediction. Different from "LM (ft)", the object decoder takes the object features right after the "Embedding layer $f_\theta$" box as input to predicate the interacted object.

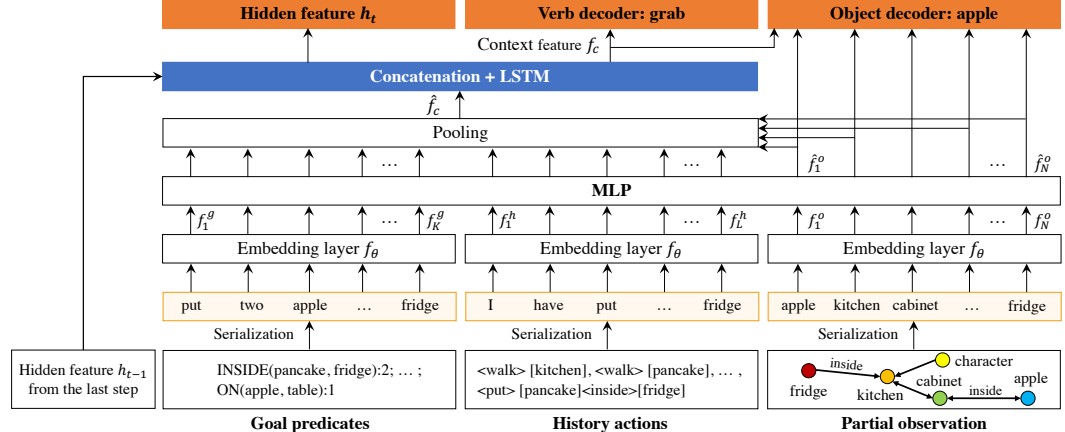

Figure A12: **Baseline LSTM.** The input and output are the same as "LM (ft)" as we introduced in the main paper Section 6 and this appendix Section A.1. "LSTM" has a similar architecture as "MLP-N" in Figure A10. The difference is in the ways to compute the context feature $f_c$. The feature $\hat{f}_c$ after the pooling layer is first concatenated with the hidden feature $h_{t-1}$ from the last step and then sent to LSTM (Hochreiter & Schmidhuber, 1997). LSTM outputs the context feature $f_c$ and a hidden feature $h_t$ that will be used for the next step. The verb decoder and object decode are the same as "MLP-N".

