# OpenReview forum: "Language Model Pre-training Improves Generalization in Policy Learning"
_ICLR.cc/2022/Conference — ICLR 2022 Submitted_

### Official Review · Reviewer_KsBt · 2021-10-30

**Correctness:** 3
**Technical Novelty And Significance:** 1
**Empirical Novelty And Significance:** 2
**Recommendation:** 3
**Confidence:** 5

**Main Review:**

Positives:
The results of language pretraining is quite large and well supported by the experimental evidence
The randomized word experiment is quite interesting and I think quite clearly demonstrates that the language pre-training is injecting information about the world and when the words are replaced by nonsense words, that knowledge can no longer be effectively transfered

Negatives:
I think the specific claim of novelty in this paper is quite weak and compared to other relevant work (which should be cited but is not) this paper is not a good contribution to the field.

Specifically, the claim is that this paper is the "First to demonstrate improved generalization in a non-linguistic problem over a standard neural-network baseline using a pre-trained language model"
Either the claim is mistaken or it is being drawn deliberately narrowly to exclude work which I think is clearly related
Specifically, I think excluding methods which have been done on text adventure games is either an accidental oversight, or this claim is just not an interesting claim of novelty. There has been quite a lot of work in this area, but I think the paper to look at that most clearly makes the claims of novelty less convincing is [1]. This paper uses a pre-trained GPT-2 model on text adventure games (specifically the  text games from the Jericho framework), also demonstrates how the pre-training improves performance, including on held-out unseen games. With [1] as context, I do not think this specific novelty claim is correct, or at best it is not an impressive claim of novelty.

This begs the question of what are the differences between this environment and text adventure environments? I would argue that in pretty much every way, this environment is less interesting.
One thing that might have been interesting is if this paper had used the 3D environment version of VirtualHome. Then I think there would be a good argument that this environment presents new challenges over text adventure games because there is an added perception problem on top of the planning / text understanding one. But since this paper exclusively used the graph-based observations and discrete symbolic actions, this is not true. I think actually that with this observation and action space it is less interesting than text world.
First, in this version, everything is templated symbols. There is a very fixed set of actions, predicates and objects that always appear written in exactly the same templated way. Contrast with text world, the space of actions is huge (essentially only limted by the original game designers imagination), there are any number of objects or other entities to interact with. The games were designed to be played by humans and are thus in much more natural language and encorporate a potentially broader set of scenarios. In VirtualHome it's just a fit set of objects and actions and the goals are just to satisfy a fixed set of predicates. And many of them are just involving the placement of various objects into other objects in a home. The fact that a symbolic planner can be used to generate ground truth (whereas you cannot do this straightforwardly in text adventure games without providing a lot of hidden information) to me indicates a less natural, less interesting setting.

In general, and specifically compared to the generalization in [1], I think the generalization in this paper is pretty limited. In this paper the test-time generalization includes novel combination of known objects in known predicates, but for instance does not include novel objects and novel predicates. There is also no novelty in actions, or in the general environment (the most is putting objects in unusual places). Contrast with [1], where entire games are held out meaning that actions, goals and objects can all be different in testing environments, making this form of generalization much more interesting.

Putting aside this specific claim of novelty, I think there are actually not a lot of new idea in this paper. The architecture is not particularly novel (just pooling the outputs of a transformer model), there isn't much novelty in training (it's just behavioral cloning) or methods, the environment is not new and as discussed is not as interesting as text games. The only real claim to novelty is that they used pretrained LMs on this specific kind of environment.

This is either a minor clarification or kind of a big problem. Is there an experiment where the raw observation/goal/actions from the environment are fed into the network COMPARED TO the transformer trained from scratch. I know there is the experiment for the first one, but not seing a comparison to a from-scratch. But in general, the confusing placement of experiments (why is the major experiment introduced and shown in the introduction?) I might have just missed this. If it's just placement, I would recommend cleaning up the experiments section. Otherwise, I think it's kind of important that the comparison is missing, because this means that the pre-training only works because the authors have essentially hand-crafted these templates to get GPT-2 to have useful pre-training.

I also found the architecture used pretty strange. Usually when people use GPT models, they will use the string token output head and just finetune it to produce the correct output. So in Figure 3, you would have all of the same inputs, but then would not generate outputs at every timestep, you would just generated a token sequence out after the last input. This gives a couple advantages:
The output heads actually contain useful training since they're trained to generate text
It lets you do zero-shot and few-shot experiments because the model can generate arbitrary string outputs. There are quite impressive results in these settings using large language models, so it is odd that this was not done.
Was there a reason why this was not done in this case? Or was this tried and found not to be as effective?

Minor:
The choice of reaching the goal in 70 steps seems somewhat arbitrary to me. Is this the default in the original environment?
ALFRED should always be in caps

[1] Keep CALM and Explore: Language Models for Action Generation in Text-based Games. Shunyu Yao, Rohan Rao, Matthew Hausknecht, Karthik Narasimhan. EMNLP 2020.

**Summary Of The Paper:**

This paper takes a transformer-based language model, pre-trained on a large text corpus (In this case GPT-2) and uses it for the symbolic version of the VirtualHome environment. The observation, goals and action history of the agent are encoded as text strings in a few various ways and fed as input to the transformer and the output of the model is pooled to predict the agent action. The paper demonstrates that in cases where the test distribution differs in some way from the training distribution, that using pre-trained transformers greatly improves performance on the task

**Summary Of The Review:**

The paper has some interesting results, but the paper lacks novelty. Pre-trained LMs have already been shown to be effective in similar text/symbolic RL environments (text adventure games) and the paper is not as impressive even as some earlier work (see [1]). It does not provide new insights/methods/architectures compared to prior literature and so I think would not be accepted.

---

### Official Review · Reviewer_g58W · 2021-11-01

**Correctness:** 4
**Technical Novelty And Significance:** 2
**Empirical Novelty And Significance:** 3
**Recommendation:** 6
**Confidence:** 5

**Main Review:**

Overall, I find the paper very well-written. It is largely empirical and the experiments are carefully setup and deliver insightful results. The claims in the introduction are supported by the experiments (but needs be further restricted to reflect the limited scope of the paper). I would like to ask whether the visual observations are used for training the models (it seems like they are not), and if not, why? In general, it'd be interesting to see whether LM-learned representation can complement other rich representations (e.g., visual representation). The symbolic, graph-based representation may be too hard for the model to learn from, and thus it is easier to observe improvement upon this representation. I suggest the authors revise the claims, emphasizing that they apply to a specific task/environment and a non-visual input representation.

**Summary Of The Paper:**

**After rebuttal**: I am keeping my score. But I will not fight against rejecting the paper. I think the results are promising but the scope of the experiments are limited and the claims need to be more precise. As pointed out in my discussion with the authors, there are also several important missing details that makes it hard to understand and appreciate the experimental settings. I like the idea of experiment 2B but the change from the string-based representation to the one-hot representation does not seem to be a significant change, as they both give the language model a sequence of word vectors. This may just show that the order of the words in the goal and history does not matter for the navigation decisions (which is kind of expected given the simple templates used to generate the string-based representation). The paper's claims would also be strengthened with reasonable explanations for the observed phenomena. The paper currently treats the experiment observations as "conclusions", which I think may be over-generalization given the limited scope of the experiments (one simulator, one type of current-state input representation).

**Before rebuttal**:
The paper proposes and studies the effectiveness of using pre-trained LMs to solve a sequential decision-making problem where the observations, goals, and actions are not originally represented in language. The authors design three experiments:
1. Convert the problem to language modeling and measure the effectiveness of pre-training the LM.
2. Compare performance of using the language input representation versus a random-string input representation.
3. Determine whether converting the inputs to texts is necessary.

The paper concludes that (1) language modeling improves generalization in policy learning, (2) language-based environment encodings are not needed to benefit from LM-pretraining, and (3) the results point the possible effectiveness of language modeling as a general-purpose pre-training scheme.



**Summary Of The Review:**

The paper offers novel, interesting results that contribute new knowledge about pretrained LMs. I recommend acceptance.

---

### Official Review · Reviewer_9PU7 · 2021-11-02

**Correctness:** 2
**Technical Novelty And Significance:** 2
**Empirical Novelty And Significance:** 2
**Recommendation:** 3
**Confidence:** 4

**Main Review:**

While the empirical results in Fig 4 do show benefits of initializing the model components from a pre-trained language model, the following are quite confusing/unclear from reading the paper:

(1) The main motivation of the paper is to explore the effect of LM pretraining on a non-linguistic task. However, the task chosen in the work is inherently linguistic -- so much so that the visual aspects of the task (i.e., observations from the environment) are represented as English tokens. Whether the tokens are structured in a formulaic representation (e.g., inside(fridge, apple)) or in traditional English sentences seems unrelated to classifying a task as linguistic v/s non-linguistic.

(2) It is unclear why the goal and history are encoded as English sentences before feeding as input to the embedding layer while the observations are input as graph entities (similar to original formulaic representation)? Is there any difference between encoding observations as graph entities v/s as English sentences? Indeed from Experiment 7.2, it is clear that there is almost negligible difference in representing goal/history/observations as English sentences vs original formulaic entities.

(3) From Fig 5, it can be observed that `LM(scratch)` is worse than `LM(ft) Random`. Given that `LM(ft) Random` is trying to learn the task using random strings, it is concerning that `LM(scratch)` is unable to outperform even that. Does this mean `LM(scratch)` is not properly tuned and/or trained till convergence?

Overall, this work presents limited empirical studies and has the above mentioned weaknesses.

**Summary Of The Paper:**

The paper studies the effect of using pre-trained components in a neural policy network for an embodied agent in a 3D environment called VirtualHome (VH). The empirical results present 3 main observations -- (1) a model with components initialized from a pre-trained language model generalized better in zero-shot settings than a model with all components randomly initialized, (2) a poorly designed (i.e., random) string encoding removed these generalization benefits, and (3) an effective encoding layer could be learned from scratch in the absence of a string-based goal and observation representation.

**Summary Of The Review:**

This work presents limited empirical studies and has several critical weaknesses.

---

### Official Review · Reviewer_NUGX · 2021-11-03

**Correctness:** 2
**Technical Novelty And Significance:** 2
**Empirical Novelty And Significance:** 2
**Recommendation:** 3
**Confidence:** 4

**Main Review:**

The authors empirically demonstrated the effectiveness of initialization with the pre-trained language model, but this result cannot give new insight that is different from existing knowledge about the effect of the pre-trained language model. Moreover, some results seem to leave concerns and questions. The detailed concerns and questions are as follows:

1. The authors assume string-based inputs (goal predicates, history actions, observation) through serialization, but I cannot agree with why it is necessary to consider this setting. Moreover, the comparison with simple baselines in Figure 4 (experiment 1) seems natural result based on existing knowledge that is not different from existing knowledge about the effectiveness of pre-trained language model.

2. For this work to be meaningful, it is thought that it should show better performance than the result performed without a pre-trained language model with original inputs that have not been serialized. However, Figure 6 (experiment 2.B) shows a negligible impact on the performance compared with learned encoding with original inputs. Based on these results, it does not sufficiently support the claim that using a pre-trained language model is helpful for generalization.

**Summary Of The Paper:**

This paper investigates the effectiveness of the language model for training the policy in embodied environments. The authors use a pre-trained GPT-2 to initialize the policy, then show the generalization effect in policy learning. In the experiments, the authors demonstrated the language model shows a better generalization effect with simple baseline and ablation studies.

**Summary Of The Review:**

This paper provides various experiments to show the generalization effect in policy learning with the pre-trained language model. However, most results cannot clearly support the main claim of the paper and it remains some concerns and questions.

---

### Decision · Program_Chairs · 2022-01-20

**Decision:**

Reject

**Comment:**

The paper studies the use of pretrained language models (LM) for training the policy in embodied environments. Specifically, a pretrained GPT-2 LM is used to initialize the policy. Environment observations, goals, and actions are encoded appropriately (e.g., converted into text strings) to apply the LM-based policy. The experiments study the generalization effect of initializing with pretrained LMs. Reviewers have found the paper made limited contributions. In particular, prior works on text adventure games have explored the use of pretrained LMs for playing games and studied the generalization effect, such as [1]. It's also suggested that the paper should revise the claims made in the experiments, given the limited experimental scope and results.

[1] Keep CALM and Explore: Language Models for Action Generation in Text-based Games. Shunyu Yao, Rohan Rao, Matthew Hausknecht, Karthik Narasimhan. EMNLP 2020.